# Legendre-KAN : High Accuracy KA Network Based on Legendre Polynomials

## Abstract

Recently, the Kolmogorov-Arnold Network (KAN) has been proposed, significantly outperforming MLP in terms of interpretability and symbolic representation. In practice, KANs are required to fit data to extremely high precision. For instance, in typical applications of KAN like inferring precise equations from data and serving as solvers for partial differential equations, high accuracy is an intrinsic requirement. In the current architecture of KAN, cubic B-spline basis functions were selected as the approximate tools. However, the inflexibility of fixed degree and knots in B-splines restricts the adaptability of the activation functions. Due to these inherent limitations of B-spline functions, especially low-order and homogeneity, KAN still has room for improvement in accuracy. In this paper, we propose the Legendre-KAN that can enhance the degrees of freedom of the basis functions in the KAN. Compared to the traditional Spline-KAN, Legendre-KAN utilizes parameterized Legendre basis functions and normalization layers at the edges of the KAN. Benefiting from higher-order orthogonal polynomials, Legendre-KAN significantly outperforms the Spline-KAN in terms of accuracy. Extensive experiments demonstrate that Legendre-KAN achieves higher accuracy and parameter efficiency, of which accuracy reaches 10 times that of Spline-KAN in some cases. For those functions which can be symbolized, this leads to more correct results as opposed to Spline-KAN. Our approach effectively improves the accuracy of the mathematical relationships in KANs, providing a better solution for approximating and analyzing complex nonlinear functions.

## 1 Introduction

Traditional neural networks have demonstrated outstanding performance in many typical fields, including image processing (Krizhevsky et al., 2012), speech recognition (Hinton et al., 2012), and natural language processing (Nadkarni et al., 2011). During the past decade, they have also become important tools for solving scientific research problems, in which the interpretability and accuracy of the network is very important. However, neural networks based on multilayer perceptrons (MLPs) are often viewed as black-box models, significantly limiting their utility in scientific research (Makke & Chawla, 2024). On the other hand, traditional activation functions, such as ReLUs, are lack of degrees of freedom and that the determination of their number and positioning is also part of the problems. A substantial number of redundant parameters are often required to achieve high-precision approximations of scientific formulas.

Recently, a new interpretable Kolmogorov-Arnold network (KAN) has been proposed. As a promising alternative to the MLP in neural networks, KAN uses parameterized B-spline basis functions at the edges of network, known as Spline-KAN. In Spline-KAN, complex multi-dimensional functions are first decomposed into the sum and composition of one-dimensional functions. These one-dimensional functions are then fitted by B-spline basis functions through training neural network and are represented as symbolic functions wherever possible. Compared to symbolic regression in traditional machine learning, Spline-KAN provides more continuous and stable results in symbolic representation tasks (Liu et al., 2024).

In scientific research based on AI, equations inferred from noise-free data demand extremely high precision. For example, the physics-informed neural networks (PINNs), which serve as numerical solvers for partial differential equations (PDEs), inherently require high accuracy (Cuomo et al.,

2022). More importantly, the accuracy of network directly impacts its reliability, which severely limits its practical applications (Korns, 2011). Especially in KAN, inaccurate predictions lead to mathematical expressions that significantly differ from the true function. The initial results of the activation functions in KAN are typically obtained through a pre-training approach. Subsequently, the functions in the set are translated and scaled to identify a symbolic function that closely aligns with the training outcomes, which serves as the mathematical expression of this activation function. However, low-precision pre-training results of Spline-KAN hinder the network's ability to find the correct symbolic representation, severely impacting the subsequent training phase and leading to a decrease in the network's accuracy. Furthermore, this may result in an analytical expression that is entirely different from the original function.

The low accuracy of Spline-KAN can be attributed to the use of B-spline basis functions. As a fundamental approximate tools, B-splines endow the Spline-KAN with superior interpretability and performance on scientific tasks compared with MLP. In traditional function approximation, low-degree piecewise polynomials in B-spline functions often fail to achieve effective fitting in certain regions of one-dimensional functions due to lower degrees of freedom. Unfortunately, the same issue persists for Spline-KAN when dealing with multidimensional functions. This implies that, in the field of symbolic representation, Spline-KAN struggles to achieve high-accuracy results with a limited number of parameters in some cases.

As a system of orthogonal polynomials defined over a finite interval, Legendre polynomials are widely used in solving differential equations, physical problems and computer graphics (Mall & Chakraverty, 2016; Parand & Razzaghi, 2004; Wu et al., 2020). When used as activation functions, Legendre polynomials can flexibly approximate complex signals with high-order and global polynomials. To improve the accuracy of traditional KAN, we proposed Legendre-KAN. This network significantly enhance the accuracy of the pre-training results, allowing for more precise symbolic approximations. In cases where certain Spline-KAN fits yield erroneous outcomes, the use of Legendre basis functions enables the Legendre-KAN to still produce relatively accurate analytical expressions. To alleviate the issue of gradient explosion commonly encountered with polynomial activation functions in neural networks, as well as the limitations imposed by the domain of Legendre polynomials, we draw inspiration from Spline-KAN and employ a periodic Min-Max function to constrain the range of input node data. This normalization layer dynamically normalizes the input data by periodically assessing the range of each input node's values. Furthermore, we identify that the gradient explosion in KAN arises from excessively large coefficients during affine transformations of the basis functions. To address this, we utilize smaller initialization coefficients in Legendre-KAN to ensure that the activation function transitions from a more stable state to the target function. Additionally, we enhance the loss function in Spline-KAN to restrict the coefficients of the basis functions within a specified range.

This paper proposes the Legendre-KAN based on the Legendre polynomials and the KA theorem. By using parameterized Legendre basis functions on the edges of the KAN, the advantages of Legendre polynomials are well integrated into the KAN. With the same number of parameters and greater time efficiency, Legendre-KAN outperforms both Spline-KAN and MLP in most functions, as illustrated in Figure 1, especially excelling in fitting complex signals. This significantly enhances the reliability and interpretability of mathematical relationships for KAN in symbolic representation tasks, which has important implications for the application and development of KA networks in natural science research.

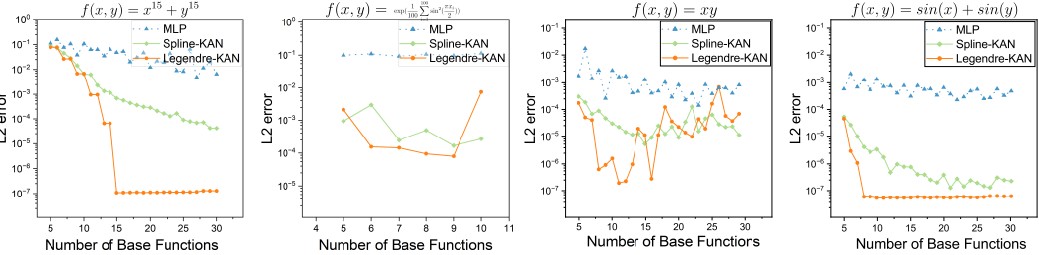

Figure 1: Test loss(RMSE) of Legendre-KAN, Spline-KAN and MLP. The high reduction rate of error shows the advantages of legendre polynomials' orthogonality.

## 2 BACKGROUND AND MOTIVATION

### 2.1 KOLMOGOROV-ARNOLD THEOREM AND KAN

The Kolmogorov-Arnold representation theorem (Arnold, 2009; Kolmogorov, 1957; Braun & Griebel, 2009) has the following general form:

$$f(\boldsymbol{x}) = f(x_1, \cdots, x_n) = \sum_{q=1}^{2n+1} \Phi_q \left( \sum_{p=1}^{n} \phi_{q,p}(x_p) \right).  \quad (1)$$

where $\Phi_q : [0,1] \to \mathbb{R}$ and $\phi_{q,p} : \mathbb{R} \to \mathbb{R}$. Its application in neural networks has been extensively studied (Lin & Unbehauen, 1993; Montanelli & Yang, 2020; Lai & Shen, 2021; He, 2023; Schmidt-Hieber, 2021). Recently, as a breakthrough application of the KA theorem in neural networks, Kolmogorov-Arnold Network(KAN) was proposed to implement a multi-level deep KA neural network, effectively combining the strengths of MLP and the KA theorem. By utilizing parameterizable activation functions, the KAN has demonstrated outstanding performance across various domains. KAN has been integrated into transformers, achieving significant success (Genet & Inzirillo, 2024). Research has also explored its application in graph neural networks, in which KAN is better than MLP (Bresson et al., 2024). Especially in AI+Science tasks, including solving differential equations and symbolic representation of functions, KAN outperforms MLPs. The great potential of KAN in scientific tasks means that further research on KAN is of considerable importance. Most critically, the KAN leverages the KA theorem to decompose the traditional multivariate function approximation problem into a composition of univariate function approximations. This implies that the accuracy or interpretability of the KAN directly depends on the univariate basis functions it employs.

### 2.2 B-SPLINE BASIS FUNCTIONS AND LEGENDRE POLYNOMIALS

In this section, we analyze the commonly used B-spline basis functions and Legendre polynomials in univariate function approximation. We first introduce the definitions of the basis functions, and then, expound our motivation for proposing Legendre-KAN.

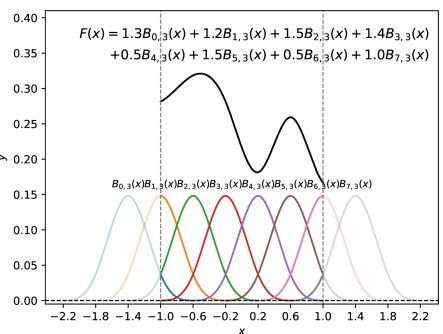 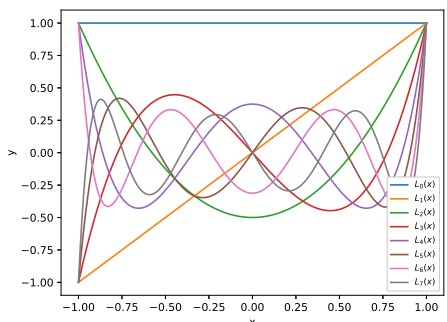

(a) Fitting result of $F(x)$, using B-spline of order 3 and 5 knots. The symbol $B_{i,k}$ means the $i^{th}$ B-spline functions of order $k$.

(b) Legendre polynomials of degree 7. The symbol $L_i(x)$ means the $i^{th}$ Legendre functions.

Figure 2: B-spline basis functions and Legendre polynomials. In each interval, only 4 cubic spline basis functions are used to fit the signal for the interval. In contrast, Legendre polynomials of degrees from 0 to 7 are used in fitting tasks.

**B-spline functions and its fitting characteristic.** A spline function is a type of function that is piecewise smooth and has a certain degree of smoothness at the junction of each piece. The term spline comes from the tool used by engineering draftsmen to connect specified points into a smooth curve. As the basic part of KAN, B-spline activation functions are the main source of its interpretability and advantages (Shukla et al., 2024), which is called Spline-KAN. The B-spline function used by KAN is the same as its form in function fitting (De Boor, 1972). For instance, the analytical

form of cubic B-splines functions can be expressed as:

$$B(x) = \begin{cases} \frac{1}{2}|x|^3 - |x|^2 + \frac{2}{3}, & |x| \leq 1 \\ -\frac{1}{6}|x|^3 + |x|^2 - 2|x| + \frac{4}{3}, & 1 < |x| \leq 2 \\ 0, & x \in others \end{cases} \quad (2)$$

In numerical analysis, a B-spline function is characterized by minimal support relative to its degree, smoothness and a specified partition of the domain (Prautzsch et al., 2002). Advantages such as locality and smoothness makes it widely used in computer graphics and computer-aided design (CAD). In function fitting, as shown in Figure 2a, function $F(x)$ is partitioned into intervals and fitted using B-spline functions.

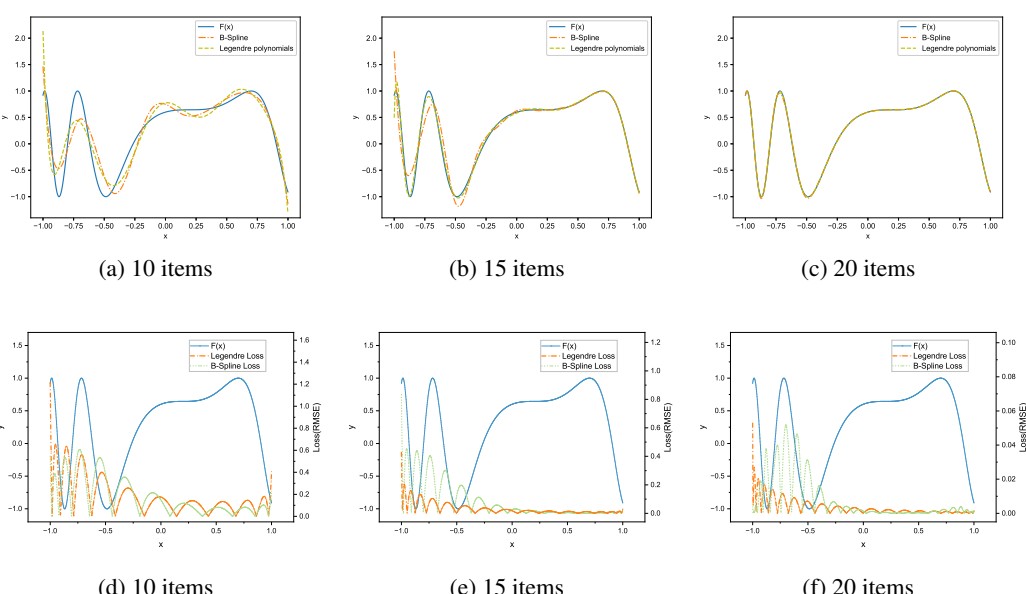

(a) 10 items      (b) 15 items      (c) 20 items

(d) 10 items      (e) 15 items      (f) 20 items

Figure 3: Fitting results and error for Legendre functions and B-spline functions. We use a function with jump and smooth regions. Each column shows the fitting results and error when using the first $n$ basis functions. The first row represents the fitting results, and the second row represents the fitting error corresponding to the results.

From a mathematical perspective, the optimal parametrized activation function trained in KAN can be interpreted as a univariate function approximation problem. As the foundation of Spline-KAN, B-spline basis functions exhibit piecewise smooth characteristics, but they are only suitable for fitting smooth, simple functions. For more complex functions, spline functions perform well in smooth regions but may introduce significant errors in jump areas, as illustrated in Figure 3. The core issue is the low degree of freedom of piecewise basis in the function approximation space of splines. The jump areas of data usually require many and higher-order basis functions to achieve high accuracy fitting, but in Spline-KAN, only a very small number of lower-order polynomials are used to fit each areas, which are not sufficient to fit the drastic changes in the jump region. This problem can severely impact the accuracy of the activation function approximation between certain nodes. Furthermore, it can degrade the overall fitting accuracy of the entire network, as shown in Figure 6. In order to improve the accuracy of the network, it is crucial to choose a basis function that has global approximation space and is higher-order stable.

**Legendre Polynomials.** Polynomial function is a class of non-linear function, recently it has also been used in neural networks, where the output is expressed as a polynomial of the input. As a special class of polynomials, Legendre polynomials were obtained by Gram-Schmidt orthogonalization of the linearly independent function system (Legendre, 1785). Rodrigul's formula gives a more concise form:

$$L_n(x) = \frac{1}{2^n n!} \frac{d^n}{dx^n}[(x^2 - 1)^n], n = 1, 2, ... \quad (3)$$

These polynomials in Figure 2b possess excellent properties such as orthogonality, even-odd symmetry and completeness. As orthogonal polynomials on a finite interval, Legendre polynomials are widely used in solving differential equations and physical problems (Al-Shaher & Mechee, 2021). Due to their excellent numerical stability and orthogonality, Legendre polynomials are also extensively applied in high-precision numerical approximation and scientific research (Ma et al., 2007). We use the recursive form of the Legendre polynomial to ensure the numerical stability when $x \in [-1, 1]$ (Abramowitz & Stegun, 1972):

$$L_n(x) = \frac{2n-1}{n} x L_{n-1}(x) - \frac{n-1}{n} L_{n-2}(x), n \geq 2 \tag{4}$$

while $L_0(x) = 1$ and $L_1(x) = x$. Recently, many studies have integrated Legendre polynomials into neural networks to achieve better results. For instance, FiLM employs Legendre polynomials to approximate historical information in time series, improving the accuracy of long-term predictions (Zhou et al., 2022). In LMU, Legendre polynomials are used to map phase space, resulting in a new high-performance recurrent neural network (Voelker et al., 2019).

As a type of basis, Legendre polynomials have a global function approximation space. Functions of different orders are all used to fit complex patterns and relationships of data, which means it has more degrees of freedom compared to B-spline basis functions. In traditional function fitting, for jump regions where B-spline fitting exhibits lower accuracy, Legendre polynomials, with their higher-degree global nature, provide a smaller and more evenly distributed approximation error, as shown in Figure 3. Based on these advantages, we propose the Legendre-KAN. Experimental results demonstrate that the aforementioned characteristics of Legendre polynomials in function fitting are well integrated into KAN, enabling Legendre-KAN to achieve 10 to 100 times higher fitting accuracy than Spline-KAN in many complex functions in Table 7. Moreover, the degree of the polynomial can be adjusted, which provides flexibility in the capacity of model.

## 3 METHODOLOGY

In this work, we propose Legendre-KAN as improvement over the accuracy of Spline-KAN. Building on the strengths of the KAN, particularly its interpretability and symbolic representation capabilities, Legendre-KAN enhance the accuracy of KAN through input normalization, reduced initialization parameters, and the incorporation of orthogonal high-order Legendre bases. We first explain the advantage of Legendre-KAN in Section 3.1. On the other hand, the structure of Legendre-KAN is presented in Section 3.2.

### 3.1 LEGENDRE-KOLMOGOROV-ARNOLD NETWORK (LEGENDRE-KAN).

As discussed below, the activation functions with lower degrees of freedom prevents Spline-KAN from producing accurate results. Local low-order basis functions struggle to achieve high-accuracy results, but as we increases the order of B-spline, test loss get worse in Appendix B. In some cases this may cause the network to choose the wrong symbolic function. In Table 2, we use KAN to approximate $F(x) = (x_0 - 10e^{-5})^{-1} + (x_1 - 10e^{-5})^{-1}$. Due to the lower accuracy of the pre-training step, Spline-KAN selects the wrong sign function on edge 1, which ultimately leads to the result which has nothing to do with the original function.

To improve the degrees of freedom of basis functions, we initially used power basis functions. Within the polynomial function space, this approach offers higher-order and global polynomials compared to the spline function space. However, these functions exhibits instability when dealing with complex tasks in Table 7, which means that power basis is not suitable for KAN. Naturally, we consider the orthogonalized version of $x^n$, the Legendre polynomials. Legendre polynomials mitigate the numerical instability inherent in power bases by applying appropriate orthogonalization. For given B-spline basis functions with $G$ knots and $K$th order, the $(K + G - 1)^{th}$ order Legendre polynomials achieve significantly higher accuracy in global function fitting compared to B-splines, as the function space contains global higher-order polynomials. With normalization and smaller initial coefficients, Legendre-KAN reduces the impact of the increase of function's order on the results of KAN. Motivated by this, we propose the Legendre-KAN.

By using parameterized Legendre functions on the edges of the KAN, the advantages of Legendre polynomials are effectively embedded into Legendre-KAN. With the same number of parameters

and improved computational efficiency, Legendre-KAN demonstrates superior accuracy in training most functions compared to Spline-KAN, particularly excelling in fitting high-order signals. This significantly enhances the reliability and interpretability of mathematical relationships in symbolic representation tasks within the KAN network, and holds substantial significance for the application and development of KAN in natural science research.

## 3.2 Architecture of Legendre-KAN

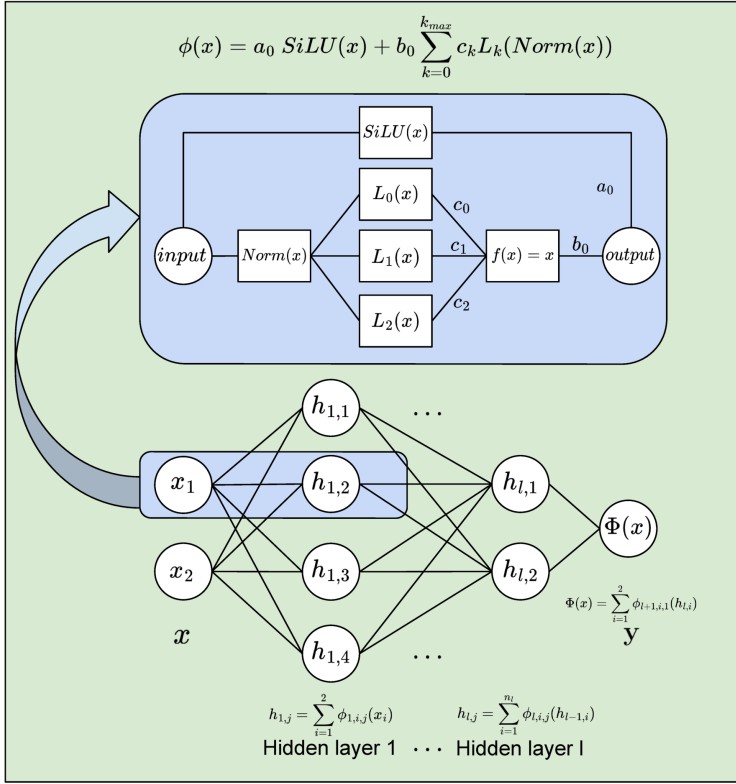

Figure 4: Architecture of Legendre-KAN, of which shape is $[2, 4, \cdots, 2, 1]$ and order is 2.

In this section we will introduce in detail the components and functions of Legendre-KAN. The entire network structure of Legendre-KAN can be found in Figure 4.

**Kolmogorov-Arnold networks.** The structure of KA networks is very different from that of MLPs. The shape of a KAN with $L$ layers can be summarized in the following form:

$$[n_1, n_2, ..., n_L], \tag{5}$$

where the number of nodes in the $i^{th}$ layer is denoted as $n_i$. The activation values of neurons in the $l^{th}$ layer and $(l + 1)^{th}$ layer can be denoted as:

$$x_{l+1,j} = \sum_{i=1}^{n_l} \phi_{l,i,j}^B(x_{l,i}), \qquad j = 1, ..., n_{l+1}$$

$$\phi_{l,i,j}^B(x) = a_{l,i,j} \, SiLU(x) + b_{l,i,j} \sum_{k=0}^{g+n} c_{l,i,j,k} B_{k,n}(x) \tag{6}$$

where $x_{l,j}$ is the $j^{th}$ node from left to right in the $l^{th}$ layer. The activation function between $x_{l+1,j}$ and $x_{l,i}$ is represented as $\phi_{l,i,j}^B$, and the $B_{k,n}(x)$ is the $k^{th}$ B-spline functions of order $n$ and $g$ knots.

Finally, the output of the network can be written in matrix form:

$$\mathbf{\Phi}_l = \begin{bmatrix} \phi_{l,1,1} & \phi_{l,1,2} & \cdots & \phi_{l,1,n_l} \\ \phi_{l,2,1} & \phi_{l,2,2} & \cdots & \phi_{l,2,n_l} \\ \vdots & \vdots & \vdots & \vdots \\ \phi_{l,n_{l+1},1} & \phi_{l,n_{l+1},2} & \cdots & \phi_{l,n_{l+1},n_l} \end{bmatrix}, \tag{7}$$

which means the output values of KAN are denoted as:

$$KAN(\boldsymbol{x}) = [\mathbf{\Phi}_L \circ \mathbf{\Phi}_{L-1} \circ ... \mathbf{\Phi}_0]\boldsymbol{x}. \tag{8}$$

**Activation Function of Legendre-KAN.** To incorporate the superior function approximation properties of Legendre polynomials into the KAN, the Legendre-KAN architecture replaces the spline activation functions used in Spline-KAN with Legendre polynomials defined on the interval [-1,1]. These Legendre polynomials are then combined linearly to form the activation functions within the network:

$$Legendre(x) = \sum_{i=0}^{k_{max}} c_i L_i(x), \tag{9}$$

where $L_i(x)$ represents the $i^{th}$ Legendre polynomials of order $k_{max}$. Since the gradient descent process essentially scales the Legendre basis functions, overly large initialization parameters can result in excessively large positive and negative coefficients for the basis functions. We use smaller initialization parameters $c_i$ to solve the problem of gradient explosion.

As a solution to the problem of low individual accuracy caused by node activation values exceeding the basis function interval, Legendre-KAN use the normalization layer to normalize the input activation values:

$$Norm(x) = \frac{2x - (x_{max} + x_{min})}{x_{max} - x_{min}}, \tag{10}$$

where $x_{max}$ and $x_{min}$ are calculated from the range of input data dynamically.

For the activation function $\phi^L(x)$ within the network, we also combined the best-performing $b(x) = SiLU(x)$ with the Legendre basis to enhance the smoothness of the high-order polynomial fitting results. To sum up, the activation function of the Legendre-KAN can be expressed as:

$$\phi^L(x) = a\, SiLU(x) + b\, Legendre(Norm(x)) \tag{11}$$

**Loss Function of Legendre-KAN.** To mitigate the effects of gradient explosion in polynomial activation functions, the loss function in Legendre-KAN incorporates a penalty term for the coefficients, which differs from Spline-KAN, as follows:

$$\mathcal{L} = \frac{1}{N_p} \sum_{i=1}^{N_p} (y_i - \hat{y}_i)^2 + \frac{\lambda}{N_c} \sum_{j=1}^{N_c} {c_j}^2 \tag{12}$$

## 4 EXPERIMENTS

In this chapter, we evaluate the accuracy and performance of the Legendre-KAN in symbolic representation tasks. Additionally, we demonstrate the high accuracy of Legendre-KAN in fitting complex nonlinear functions that cannot be symbolized. Furthermore, we visualize the result errors to demonstrate how high-order polynomial bases enhance the ability of KAN in capturing complex global signals. Finally, we conduct ablation experiments to assess the impact of each component on the performance of Legendre-KAN. The results of other KANs, including Fourier-KAN and Wavlet-KAN, are in Appendix D.2 (Xu et al., 2024; Bozorgasl & Chen, 2024).

**Experimental Setup.** All functions tested in this study are sourced from the KAN and Feynman datasets (Udrescu et al., 2020; Udrescu & Tegmark, 2020). For all experiments, original input is normalized to the range $[-1, 1]$. To ensure fairness, we compare the Legendre-KAN with k order Legendre polynomial basis functions to Spline-KAN and MLPs with the same number of parameters, as detailed in Appendix A. We perform hyperparameter tuning on the training set. Each method has a couple of hyperparameters: the learning rate, number of epochs and coefficients scale, more details are under Appendix C.

## 4.1 Performance on Symbolic Functions

Table 1: Test loss and average running time over symbolic functions

| Function | Shape | Spline-KAN | | | | Legendre-KAN | | |
|---|---|---|---|---|---|---|---|---|
| | | best k/grid | best loss | loss (equal params) | Train/epoch(s) | best k | best loss | Train/epoch(s) |
| $x_0 x_1$ | $[2,2,1]$ | 4/16 | $2.04 \times 10^{-4}$ | $4.26 \times 10^{-4}$ | 0.122494 | 7 | $\mathbf{3.48 \times 10^{-5}}$ | $\mathbf{0.066302}$ |
| $(x_0+2)/(x_1+2)$ | $[2,2,1]$ | 3/13 | $\mathbf{1.44 \times 10^{-5}}$ | $1.72 \times 10^{-4}$ | 0.121438 | 6 | $3.70 \times 10^{-5}$ | $\mathbf{0.045682}$ |
| $\sqrt{x_0+1}+\sqrt{x_1+1}$ | $[2,1]$ | 5/16 | $1.01 \times 10^{-3}$ | $1.01 \times 10^{-3}$ | 0.028822 | 20 | $\mathbf{7.29 \times 10^{-4}}$ | $\mathbf{0.016484}$ |
| $(x_0+2)^{-1}+(x_1+2)^{-1}$ | $[2,1]$ | 3/18 | $1.05 \times 10^{-6}$ | $3.16 \times 10^{-6}$ | 0.037084 | 15 | $\mathbf{8.37 \times 10^{-8}}$ | $\mathbf{0.013190}$ |
| $\sin(x_0)+\sin(x_1)$ | $[2,1]$ | 3/18 | $1.71 \times 10^{-7}$ | $5.38 \times 10^{-7}$ | 0.032387 | 15 | $\mathbf{5.78 \times 10^{-8}}$ | $\mathbf{0.013335}$ |
| $\tan(x_0)+\tan(x_1)$ | $[2,1]$ | 3/18 | $1.26 \times 10^{-5}$ | $1.83 \times 10^{-5}$ | 0.040951 | 18 | $\mathbf{1.03 \times 10^{-7}}$ | $\mathbf{0.013565}$ |
| $\arcsin(x_0)+\arcsin(x_1)$ | $[2,1]$ | 4/17 | $2.40 \times 10^{-3}$ | $2.40 \times 10^{-3}$ | 0.024183 | 20 | $\mathbf{1.66 \times 10^{-3}}$ | $\mathbf{0.012618}$ |
| $\exp(x_0)+\exp(x_1)$ | $[2,1]$ | 3/17 | $5.15 \times 10^{-7}$ | $2.70 \times 10^{-6}$ | 0.030216 | 13 | $\mathbf{1.52 \times 10^{-7}}$ | $\mathbf{0.011386}$ |
| $\log(x_0+2)+\log(x_1+2)$ | $[2,1]$ | 3/16 | $2.49 \times 10^{-7}$ | $9.81 \times 10^{-7}$ | 0.024169 | 15 | $\mathbf{7.07 \times 10^{-8}}$ | $\mathbf{0.013442}$ |

Our experiments first focus on simple fundamental functions that can be symbolically fixed. The training accuracy of these simple functions directly affects the overall accuracy and fitting precision of the network. We selected representative basis functions used in the fixed-step process of the KAN network as test signals. These simple, symbolizable functions provide a good evaluation of the performance of the Legendre-KAN in terms of symbolic representation and data fitting. As shown in Table 1, Legendre-KAN demonstrates extremely high accuracy in symbolic function representation. It achieves lower optimal loss and faster training speed, when using same parameter quantity, compared to Spline-KAN. Additionally, Legendre-KAN is able to achieve superior fitting results with fewer parameters. This is because the globality and orthogonality of the Legendre polynomials, which ameliorates the parameter efficiency of KAN.

## 4.2 Evaluation of Symbolic Representation Ability

Table 2: Fitting results of $f(x)$, including symbolic results and training loss

| Network | Edge number | Symbolic function / $r_2$ | | | Loss(RMSE) | Symbolic representation results | Loss(RMSE) |
|---|---|---|---|---|---|---|---|
| | | $1^{th}$ | $2^{th}$ | $3^{th}$ | (pre-training) | | (final training) |
| Spline-KAN | $(0,1,1)$ | $x^{-1}$ / 80.8% | $\tan(x)$ / 79.9% | $\log(x)$ / 25.7% | $8.69 \times 10^{1}$ | $1.05816(x_0+0.00011)^{-1}$ | $1.25 \times 10^{1}$ |
| | $(0,2,1)$ | $\mathbf{\tan(x)}$ / 82.2% | $x^{-1}$ / 81.6% | $\exp(x)$ / 64.2% | | $-2.61321\,\mathbf{\tan(2.09370x_1 - 1.57004)} - 12.02730$ | |
| Legendre-KAN | $(0,1,1)$ | $x^{-1}$ / 84.4% | $\tan(x)$ / 84.1% | $\log(x)$ / 17.8% | $6.11 \times 10^{1}$ | $(x_0 - 0.00010)^{-1}$ | $8.82 \times 10^{-6}$ |
| | $(0,2,1)$ | $x^{-1}$ / 80.5% | $\tan(x)$ / 80.4% | $\log(x)$ / 52.8% | | $(x_1 - 0.00010)^{-1}$ | |

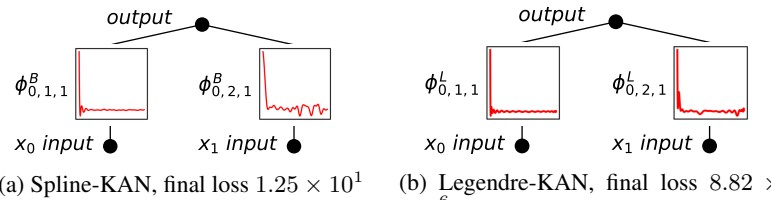

(a) Spline-KAN, final loss $1.25 \times 10^1$   (b) Legendre-KAN, final loss $8.82 \times 10^{-6}$

Figure 5: Results of $f(x) = (x_0 + 10^{-5})^{-1} + (x_1 + 10^{-5})^{-1}$ after pre-training, where $\phi_{l,i,j}(x)$ is the activation functions of edge $(l,i,j)$ between $\mathrm{node}(l,i)$ and $\mathrm{node}(l+1,j)$. The functions $\phi^B(x)$ and $\phi^L(x)$ are defined in Section 3.2

The accuracy achieved during initial training significantly impacts the correctness of the final symbolic representation. The high precision of Legendre-KAN allows it to provide a more easily fixed initial result within the KAN, leading to a final symbolic representation accuracy. As illustrated in Figure 5 and Table 2, unlike Spline-KAN, the higher accuracy enables Legendre-KAN to fit function which requires precise modeling. After setting and next training, Legendre-KAN gets true result accurate to five decimal places, but Spline-KAN has an error on $\mathrm{edge}(0,1,1)$.

As shown in Table 3, difference between Spline-KAN and Legendre-KAN is not obvious on simple functions. However, when dealing with complex coefficients, as highlighted in the table, Legendre-KAN's superior precision yields results that closely align with the original function. Importantly, for instances where Spline-KAN produces erroneous fittings, Legendre-KAN is capable of accurately

capturing the correct function, demonstrating only minor discrepancies in coefficient accuracy compared to the original function. The loss ratio represents the ratio of the RMSE between Spline-KAN and Legendre-KAN. It is evident that for functions with incorrect symbolic representations, the error difference between the two methods is significant. These experiments underscore the critical role of accuracy in KAN for both symbolic representation and interpretability.

Table 3: Results of symbolic functions

| Origin function | Result of Legendre-KAN | Result of Spline-KAN | Loss ratio |
|---|---|---|---|
| $\exp(\sin(x_0) + x_1^2)$ | $\exp(\sin(x_0) + x_1^2)$ | $\exp(\sin(x_0) + x_1^2)$ | $1.37 \times 10^0$ |
| $x_0^8 + x_1^4 + x_2^2 + x_4$ | $x_0^8 + x_1^4 + x_2^2 + x_4$ | $x_0^8 + x_1^4 + x_2^2 + x_4$ | $8.74 \times 10^1$ |
| $1.00\sin(12x_0) + 1.00\sin(12x_1)$ | $0.98\sin(11.86x_0) + 0.99\sin(11.89x_1)$ | $0.95\sin(11.74x_0 + 0.01) + 0.95\sin(11.76x_1 + 0.01)$ | $4.05 \times 10^0$ |
| $\tanh(1.00x_0^4 + 1.00x_1^4 + 1.00x_2^4 + 1.00x_3^4 + 1)$ | $1.05\tanh(0.94x_0^4 + 0.94x_1^4 + 0.94x_2^4 + 0.94x_3^4 - 0.95) + 0.01$ | $1.01\tanh(0.99x_0^4 + 0.99x_1^4 + 0.99x_2^4 + 0.99x_3^4 - 0.99)$ | $5.12 \times 10^1$ |
| $1.00\exp(\sin(x_0) + x_1^2 + \lvert 1.00x_2\rvert)$ | $1.00\exp(\sin(x_0) + x_1^2 + \lvert 0.94x_2\rvert) + 0.01$ | $1.71\exp(\sin(x_0) + x_1^2 + 0x_2)$ | $2.02 \times 10^4$ |
| $(1.00\exp(x_0) + 1.00\exp(x_1))^4$ | $1.03(1.00\exp(0.99x_0) + 0.98\exp(1.01x_1))^4 - 0.05$ | $7.11\exp(0.91\exp(x_0) + 0.91\exp(x_1)) - 24.44$ | $1.44 \times 10^3$ |
| $\log(x_0^4 + x_1^2 + 1.00)$ | $\log(x_0^4 + x_1^2 + 1.00)$ | $0.71(-0.71\tanh(5.62x_0 - 5.39) - (x_1 + 0.01)^4 - 0.68)^{-1} + 0.13$ | $7.11 \times 10^6$ |

### 4.3 PERFORMANCE ON COMPLEX NONLINEAR FUNCTIONS

In this subsection, we evaluate the performance of KANs and MLPs on complex nonlinear functions. We use $x^n$ as the activation function of Polynomial-KAN, and the number of layers in MLPs ranges from 3 to 6. In Appendix D.1, for oscillatory functions, polynomial functions and complex composite functions, Legendre-KAN achieves extremely high accuracy. Additionally, we test scaling and translation of symbolic functions. For these types of functions, changes in the input interval significantly affect the fitting performance of Spline-KAN, but Legendre-KAN still produces high-accuracy results.

In function fitting, high-order polynomials often leads to numerical instability and oscillation, which severely affects fitting accuracy. However, as shown in Table 4, high-order and high-dimensional Legendre polynomials do not compromise the fitting efficiency of Legendre-KAN.

Table 4: Test loss (RMSE) over high-dimensional and high-order functions

| Function | Spline-KAN | | Polynomial-KAN | | Legendre-KAN | | MLPs | |
|---|---|---|---|---|---|---|---|---|
| | Lowest loss | Time ratio | Lowest loss | Time ratio | Lowest loss | Time ratio | Lowest loss | Time ratio |
| $x_0^{30} + x_1^{20} + x_2^{10} + x_3 + 1$ | $2.62 \times 10^{-4}$ | 5.33 | $1.66 \times 10^{-4}$ | 1.66 | $\mathbf{1.68 \times 10^{-7}}$ | 1.00 | $2.82 \times 10^{-2}$ | **0.99** |
| $x_0^{60} + x_1^{45} + x_2^{30} + x_3^{15} + 1$ | $4.54 \times 10^{-4}$ | 2.57 | $9.69 \times 10^{-5}$ | 2.16 | $\mathbf{3.57 \times 10^{-7}}$ | 1.00 | $2.05 \times 10^{-2}$ | **0.72** |
| $\exp\left(\frac{1}{100}\sum_{i=0}^{100}\sin^2(\frac{\pi}{2}x_i)\right)$ | $4.58 \times 10^{-4}$ | 0.75 | $1.69 \times 10^{-3}$ | 2.16 | $\mathbf{8.10 \times 10^{-5}}$ | 1.00 | $8.26 \times 10^{-2}$ | **0.20** |
| $\tanh(x_0^4 + x_1^4 + x_2^4 + x_3^4 - 1)$ | $2.50 \times 10^{-4}$ | 2.44 | $9.13 \times 10^{-4}$ | 1.72 | $\mathbf{1.03 \times 10^{-4}}$ | 1.00 | $5.69 \times 10^{-2}$ | **0.26** |
| $x_0\big((x_1 - 2)^2 + (x_2 - x_3)^2 + (x_4 - x_5)^2\big)^{-1}$ | $1.89 \times 10^{-2}$ | 2.14 | $1.34 \times 10^{-2}$ | 0.59 | $5.41 \times 10^{-2}$ | 1.00 | $\mathbf{2.38 \times 10^{-3}}$ | **0.04** |

### 4.4 VISUALIZATION OF TEST LOSS

To demonstrate the impact of the Legendre polynomial basis on the accuracy of the KAN, Figure 6 visualizes the test loss for some binary functions. By comparing Figure 6a and 6e, the $l_2$ error for Spline-KAN increases especially when x approaches -1 and 1. This is because the function $f(x, c)$ cannot be effectively captured by low-order spline bases when x nears -1 and 1. In contrast, the presence of higher-order global polynomial bases in the Legendre polynomial function space allows the KANs to capture the global signal. As shown in Figure 6i, this results in lower and more uniformly distributed errors across the test samples.

### 4.5 ABLATION ANALYSIS

To investigate the impact of different modules on the accuracy of Legendre-KAN, we conducted ablation experiments focusing on parameter initialization, normalization functions, and basis functions. As previously discussed, Legendre-KAN shows a significant advantage over Spline-KAN

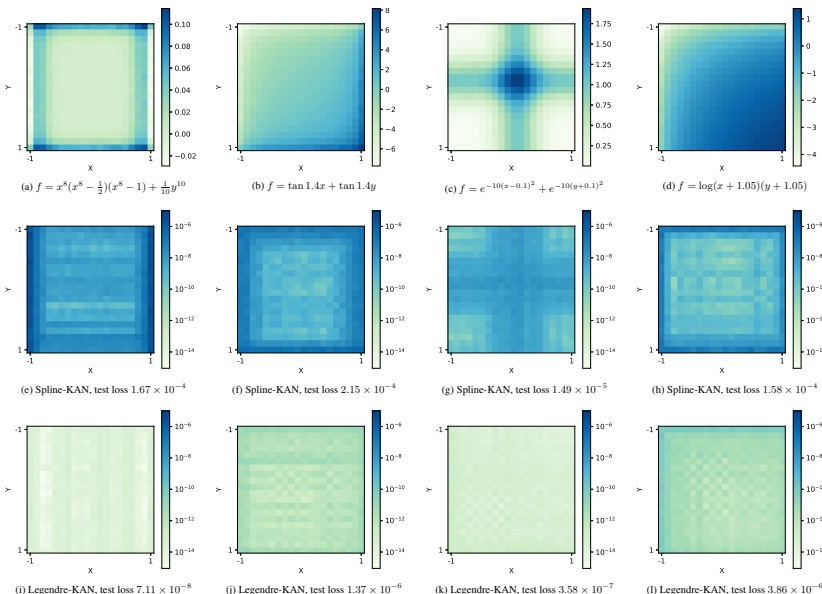

Figure 6: Visualization of test loss for Spline-KAN and Legendre-KAN

for polynomial problems. When extended to specific problems, Legendre-KAN may demonstrate higher interpretability and accuracy.

**Parameter Initialization:** We present the impact of initialization parameters on model accuracy in Table N. We evaluated the effect of initialization parameters ranging from 1 to on accuracy and gradient explosion during model training. Figure 7a shows the results of $f(x) = x_0^{15} + x_1^{10} + x_2^5 + x_3^{15} + 1$.

**Normalization Functions:** To assess the impact of normalization functions on model performance, We first tested five different normalization functions: Min-Max, DSILU, ReLU, Tanh and a combined function. The results of $f(x) = x_0^{15} + x_1^{10} + x_2^5 + x_3^1 + 1$ are shown in Figure 7b.

**Combination:** In addition, we evaluate the importance of normalization function(Min-Max) and parameter initialization in Figure 7c. The PDE function is $\nabla^2 f(x,y) = 90x^8 + 20\sin^3 y - 25\sin^5 y$, of which the truth solution is $f(x,y) = x^{10} + \sin^5 y$.

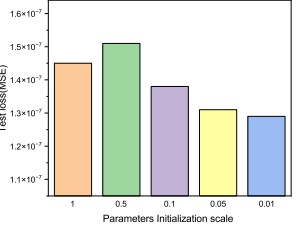 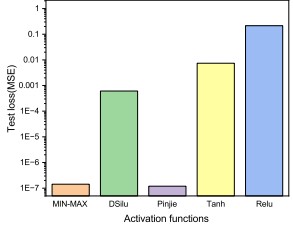 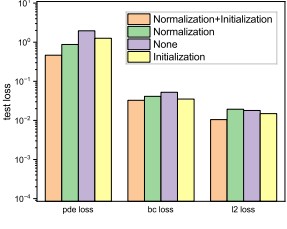

(a) Results of test function with different parameters initialization

(b) Results of test function with different activation functions

(c) Results of PDE

Figure 7: Ablation analysis

## 5 CONCLUSION

This paper proposes the Legendre-KAN that combines the advantages of Legendre polynomials and the KA theorem. This network leverages the global and orthogonal properties of the Legendre basis functions, effectively addressing the numerical approximation accuracy issues associated with low-order spline bases. Extensive experiments demonstrate that Legendre-KAN achieves higher accuracy and interpretability in symbolic representation, compared to the Spline-KAN, while using less time and parameters.

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

## A    ANALYSIS OF NETWORK PARAMETERS

For the Legendre-KAN with $k-1$-th order Legendre polynomial basis functions, and the Spline-KAN using $k'$-th order B-spline basis with $k-k'$ intervals, both networks have $k$ activation functions on their edges. The number of parameters on each edge is $k+3$, which includes the coefficients for the $k$ activation functions, two additional weighting parameters, and a bias after the final activation value.

When the structure of KAN is $[n_0, n_1, ..., n_l]$, with k activation functions on each edge, the total number of parameters can be presented as :

$$N_k = (k+3) \sum_{i=1}^{l-1} n_i n_{i+1} \tag{13}$$

When the structure of MLP is $[n_0, n_1, ..., n_l]$, the total number of parameters can be presented as :

$$N_m = \sum_{j=1}^{l-1} (n_j + 1) n_{j+1} \tag{14}$$

## B  THE EFFECT OF THE ORDER OF SPLINE-KAN

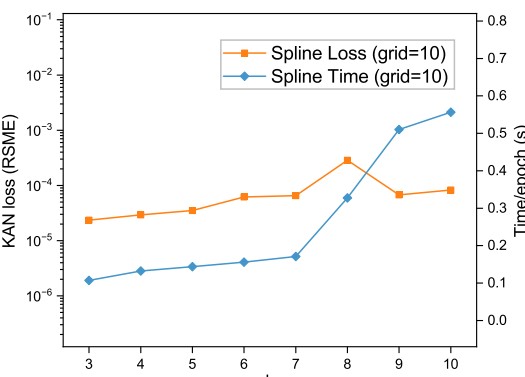

Figure 8: Test loss and training time of KAN.

In this figure, we evaluate the effect of the order of Spline-KAN when fitting $F(x) = (x_0 - 10e^{-5})^{-1} + (x_1 - 10e^{-5})^{-1}$. Surprisingly, as the order increases, the fitting results of $F(x)$ become worse. This means that increasing the order will not produce a more precise result. Therefore, the low-order B-spline basis function is used in the experiments of Spline-KAN in this paper.

## C  EXPERIMENT PARAMETER

The experiments were performed on a desktop PC with Intel Core i5-13600kf processor, 4070Ti GPU, 32G memory.

During hyperparameter tuning for KAN, including Spline-KAN and Legendre-KAN, we perform a grid search over the values $\{10, 1, 0.1, 0.5, 0.01, 0.05, 0.001, 0.0001\}$ for the learning rate, $\{10, 20, 40, 60, 80, 100\}$ for the number of epochs, $\{0.1, 0.01, 0.001, 0.0001\}$ for the initial coefficients scale. For each hyperparameter combination, we run 3 random seeds and choose the best result.

**Some explanations of indicators used in this article.** All the **ratios** represent the ratio of the indicator of a certain network to this indicator of Legendre.

For each tasks, we test the network with different parameter quantity. The parameter quantity for the lowest error of the result of the Legendre-KAN is assumed to be $k_l$. We divide all the results of networks into two parts. The **first part** is the part where the parameter quantity is less than or equal to or slightly greater than $k_l$. The **other part** is the part where the number of parameters is less than or equal to the maximum number of basis functions specified. The former is to compare the situation when the two networks' parameter quantities are equal. The second part is used to compare the optimal fitting accuracy of the network under a certain number of parameters. For the result with the lowest error in the first part, the number of basis functions is described as **Equal params**. For the result with the lowest error in the second part, the number of basis functions is described as **best k/grid** in Spline-KAN and **best k** in other networks.

The parameter quantity of the network under optimal loss:

Table 5: Test loss over complex nonlinear functions

| Function | KAN shape | Spline-KAN | | Polynomial-KAN | | Legendre-KAN | MLPs | |
|---|---|---|---|---|---|---|---|---|
| | | best k/grid | Equal params | best k | Equal params | best k | best shape | best shape |
| $x_0^7 + x_1^4 + x_2^3 + x_3 + 1$ | [4, 1] | 4/22 | 4/4 | 7 | 7 | 7 | [4, 23, 1] | [4, 7, 1] |
| $x_0^{15} + x_1^{10} + x_2^5 + x_3 + 1$ | [4, 1] | 4/25 | 4/12 | 21 | 15 | 15 | [4, 23, 1] | [4, 13, 1] |
| $x_0^8(x_0^8 - \frac{1}{2})(x_0^8 - 1) + \frac{1}{10}x_1^{10}$ | [2, 1] | 4/26 | 4/22 | 24 | 24 | 24 | [2, 8, 3, 3, 1] | [2, 7, 2, 4, 1] |
| $J_0(20x_0) + x_1^2$ | [2, 5, 1] | 5/26 | 5/11 | 14 | 14 | 15 | [2, 41, 8, 4, 1] | [2, 28, 5, 4, 1] |
| $tan(1.4x_0) + tan(1.4x_1)$ | [2, 1] | 4/27 | 4/26 | 29 | 29 | 29 | [2, 16, 1] | [2, 15, 1] |
| $log((x_0+1.05)(x_1+1.05))$ | [2, 1] | 4/27 | 4/26 | 30 | 30 | 30 | [2, 15, 1] | [2, 15, 1] |
| $exp(-10(x_0-0.1)^2) + exp(-10(x_1+0.1)^2)$ | [2, 1] | 4/27 | 4/22 | 11 | 11 | 25 | [2, 11, 2, 1] | [2, 14, 1] |
| $exp(sin(x_0) + x_1^2)$ | [2, 1, 1] | 3/24 | 4/14 | 6 | 6 | 17 | [2, 11, 4, 3, 1] | [2, 7, 3, 3, 1] |
| $(x_0 + x_1)/(2 + x_0x_1)$ | [2, 2, 1] | 3/23 | 4/14 | 9 | 9 | 14 | [2, 18, 7, 2, 1] | [2, 12, 4, 3, 1] |

Table 6: Test loss over high-dimensional and high-order functions

| Function | KAN shape | Spline-KAN | Polynomial-KAN | Legendre-KAN | MLPs |
|---|---|---|---|---|---|
| | | best k/grid | best k | best k | best shape |
| $x_0^{30} + x_1^{20} + x_2^{10} + x_3 + 1$ | [4, 1] | 4/27 | 30 | 30 | [4, 20, 1] |
| $x_0^{60} + x_1^{45} + x_2^{30} + x_3^{15} + 1$ | [4, 1] | 4/57 | 60 | 60 | [4, 15, 7, 1] |
| $exp\left(\frac{1}{100}\sum_{i=0}^{100} sin^2(\frac{\pi}{2}x_i)\right)$ | [100, 1, 1] | 3/6 | 7 | 8 | [100, 9, 1, 1] |
| $tanh(x_0^4 + x_1^4 + x_2^4 + x_3^4 - 1)$ | [4, 1, 1] | 6/12 | 15 | 17 | [4, 13, 1] |
| $x_0((x_1 - 2)^2 + (x_2 - x_3)^2 + (x_4 - x_5)^2)^{-1}$ | [6, 4, 1, 1] | 6/25 | 9 | 30 | [6, 13, 4, 2, 1] |

# D EXPERIMENT DETAILS OF COMPLEX FUNCTIONS

## D.1 EXPERIMENT DETAILS OF NONLINEAR FUNCTIONS

Table 7: Test loss over complex nonlinear functions

| Function | Spline-KAN | | Polynomial-KAN | | Legendre-KAN | MLPs | |
|---|---|---|---|---|---|---|---|
| | Lowest loss | Equal params | Lowest loss | Equal params | Lowest loss | Lowest loss | Equal params |
| | | | | Test loss (RMSE) | | | |
| $x_0^7+x_1^4+x_2^3+x_3+1$ | $8.19 \times 10^{-6}$ | $1.48 \times 10^{-3}$ | $2.51 \times 10^{-6}$ | $2.51 \times 10^{-6}$ | $\mathbf{8.03 \times 10^{-7}}$ | $2.48 \times 10^{-3}$ | $3.58 \times 10^{-2}$ |
| $x_0^{15}+x_1^{10}+x_2^5+x_3+1$ | $5.15 \times 10^{-5}$ | $4.44 \times 10^{-4}$ | $1.00 \times 10^{-4}$ | $1.25 \times 10^{-4}$ | $\mathbf{1.29 \times 10^{-7}}$ | $4.48 \times 10^{-3}$ | $2.08 \times 10^{-2}$ |
| $x_0^8(x_0^8 - \frac{1}{2})(x_0^8 - 1) + \frac{1}{10}x_1^{10}$ | $1.05 \times 10^{-4}$ | $1.67 \times 10^{-4}$ | $8.09 \times 10^{-5}$ | $8.09 \times 10^{-5}$ | $\mathbf{7.11 \times 10^{-8}}$ | $3.43 \times 10^{-3}$ | $1.88 \times 10^{-2}$ |
| $J_0(20x_0)+x_1^2$ | $2.21 \times 10^{-4}$ | $2.47 \times 10^{-4}$ | $7.82 \times 10^{-4}$ | $7.82 \times 10^{-4}$ | $\mathbf{6.49 \times 10^{-5}}$ | $4.63 \times 10^{-3}$ | $5.02 \times 10^{-2}$ |
| $tan(1.4x_0) + tan(1.4x_1)$ | $2.15 \times 10^{-4}$ | $2.15 \times 10^{-4}$ | $5.61 \times 10^{-4}$ | $5.61 \times 10^{-4}$ | $\mathbf{1.37 \times 10^{-6}}$ | $1.44 \times 10^{-2}$ | $1.48 \times 10^{-2}$ |
| $log(x_0+1.05) + log(x_1+1.05)$ | $8.77 \times 10^{-5}$ | $1.58 \times 10^{-4}$ | $3.87 \times 10^{-4}$ | $3.87 \times 10^{-4}$ | $\mathbf{3.86 \times 10^{-6}}$ | $2.82 \times 10^{-3}$ | $2.82 \times 10^{-3}$ |
| $exp(-10(x_0-0.1)^2) + exp(-10(x_1+0.1)^2)$ | $1.03 \times 10^{-5}$ | $1.49 \times 10^{-5}$ | $8.53 \times 10^{-3}$ | $8.53 \times 10^{-3}$ | $\mathbf{3.58 \times 10^{-7}}$ | $9.27 \times 10^{-4}$ | $3.24 \times 10^{-3}$ |
| $exp(sin(x_0) + x_1^2)$ | $9.91 \times 10^{-6}$ | $1.01 \times 10^{-5}$ | $3.37 \times 10^{-5}$ | $3.37 \times 10^{-5}$ | $\mathbf{4.59 \times 10^{-7}}$ | $2.52 \times 10^{-4}$ | $3.33 \times 10^{-4}$ |
| $(x_0 + x_1)/(2 + x_0x_1)$ | $3.81 \times 10^{-5}$ | $4.88 \times 10^{-5}$ | $9.23 \times 10^{-5}$ | $9.23 \times 10^{-5}$ | $\mathbf{1.88 \times 10^{-5}}$ | $1.57 \times 10^{-4}$ | $2.90 \times 10^{-4}$ |
| | | | | Time ratio (compared with Legendre-KAN) | | | |
| $x_0^7+x_1^4+x_2^3+x_3+1$ | 6.24 | 4.79 | 1.68 | 1.68 | 1.00 | 0.89 | **0.62** |
| $x_0^{15}+x_1^{10}+x_2^5+x_3+1$ | 4.07 | 5.18 | 1.99 | 2.14 | 1.00 | **1.00** | 1.09 |
| $x_0^8(x_0^8 - \frac{1}{2})(x_0^8 - 1) + \frac{1}{10}x_1^{10}$ | 3.03 | 2.18 | 1.58 | 1.58 | 1.00 | 0.41 | **0.37** |
| $J_0(20x_0)+x_1^2$ | 2.31 | 1.55 | 1.02 | 1.02 | 1.00 | 0.11 | **0.10** |
| $tan(1.4x_0) + tan(1.4x_1)$ | 2.76 | 2.76 | 1.61 | 1.61 | 1.00 | **0.63** | 0.82 |
| $log(x_0+1.05) + log(x_1+1.05)$ | 3.29 | 2.42 | 1.47 | 1.47 | 1.00 | 0.63 | **0.63** |
| $exp(-10(x_0-0.1)^2) + exp(-10(x_1+0.1)^2)$ | 2.80 | 2.37 | 1.82 | 1.82 | 1.00 | 0.85 | **0.80** |
| $exp(sin(x_0) + x_1^2)$ | 1.47 | 1.47 | 0.92 | 0.06 | 1.00 | 0.24 | **0.24** |
| $(x_0 + x_1)/(2 + x_0x_1)$ | 1.92 | 1.76 | 1.63 | 1.63 | 1.00 | 0.21 | **0.20** |

## D.2 EXPERIMENT DETAILS OF COMPLEX FUNCTIONS WHEN USING OTHER KANS

Table 8: Test loss and average running time over symbolic functions

| Function | Shape | Fourier-KAN | | | | Wavlet-KAN | | | Legendre-KAN | | |
|---|---|---|---|---|---|---|---|---|---|---|---|
| | | best k | best loss | loss (equal params) | Train/epoch(s) | best k | best loss | Train/epoch(s) | best k | best loss | Train/epoch(s) |
| $x_0 x_1$ | [2,2,1] | 6 | $1.18 \times 10^{-3}$ | $2.26 \times 10^{-3}$ | – | $3.18 \times 10^{-3}$ | $6.05 \times 10^{-3}$ | – | 7 | $\mathbf{3.48 \times 10^{-5}}$ | 0.066302 |
| $(x_0 + 2)/(x_1 + 2)$ | [2,2,1] | 11 | $8.46 \times 10^{-4}$ | $2.17 \times 10^{-3}$ | – | $1.39 \times 10^{-3}$ | $2.54 \times 10^{-3}$ | – | 6 | $\mathbf{3.70 \times 10^{-5}}$ | 0.045682 |
| $\sqrt{x_0 + 1} + \sqrt{x_1 + 1}$ | [2,1] | 18 | $6.61 \times 10^{-4}$ | $2.63 \times 10^{-3}$ | – | $2.28 \times 10^{-3}$ | $2.28 \times 10^{-3}$ | – | 20 | $\mathbf{7.29 \times 10^{-4}}$ | 0.016484 |
| $(x_0 + 2)^{-1} + (x_1 + 2)^{-1}$ | [2,1] | 19 | $1.70 \times 10^{-5}$ | $7.13 \times 10^{-5}$ | – | $1.05 \times 10^{-3}$ | $1.45 \times 10^{-3}$ | – | 15 | $\mathbf{8.37 \times 10^{-8}}$ | 0.013190 |
| $\sin(x_0) + \sin(x_1)$ | [2,1] | 12 | $7.18 \times 10^{-5}$ | $1.75 \times 10^{-4}$ | – | $9.39 \times 10^{-4}$ | $1.04 \times 10^{-3}$ | – | 15 | $\mathbf{5.78 \times 10^{-8}}$ | 0.013335 |
| $\tan(x_0) + \tan(x_1)$ | [2,1] | 18 | $9.23 \times 10^{-4}$ | $1.11 \times 10^{-3}$ | – | $2.49 \times 10^{-3}$ | $2.49 \times 10^{-3}$ | – | 18 | $\mathbf{1.03 \times 10^{-7}}$ | 0.013565 |
| $\arcsin(x_0) + \arcsin(x_1)$ | [2,1] | 15 | $2.00 \times 10^{-5}$ | $2.00 \times 10^{-5}$ | – | $5.13 \times 10^{-3}$ | $5.13 \times 10^{-3}$ | – | 20 | $\mathbf{1.66 \times 10^{-3}}$ | 0.012618 |
| $\exp(x_0) + \exp(x_1)$ | [2,1] | 15 | $2.21 \times 10^{-4}$ | $2.87 \times 10^{-4}$ | – | $6.14 \times 10^{-4}$ | $6.14 \times 10^{-4}$ | – | 13 | $\mathbf{1.52 \times 10^{-7}}$ | 0.011386 |
| $\log(x_0 + 2) + \log(x_1 + 2)$ | [2,1] | 13 | $7.76 \times 10^{-5}$ | $9.80 \times 10^{-5}$ | – | $8.83 \times 10^{-4}$ | $1.54 \times 10^{-3}$ | – | 15 | $\mathbf{7.07 \times 10^{-8}}$ | 0.013442 |

Table 9: Test loss over complex nonlinear functions

| Function | Fourier-KAN | | Wavlet-KAN | | Legendre-KAN |
|---|---|---|---|---|---|
| | Lowest loss | Equal params | Lowest loss | Equal params | Lowest loss |
| | Test loss (RMSE) | | | | |
| $x_0^7 + x_1^4 + x_2^3 + x_3 + 1$ | $3.24 \times 10^{-4}$ | $3.86 \times 10^{-3}$ | $3.04 \times 10^{-3}$ | $7.11 \times 10^{-3}$ | $\mathbf{8.03 \times 10^{-7}}$ |
| $x_0^{15} + x_1^{10} + x_2^5 + x_3 + 1$ | $6.25 \times 10^{-4}$ | $2.80 \times 10^{-3}$ | $4.65 \times 10^{-3}$ | $9.45 \times 10^{-3}$ | $\mathbf{1.29 \times 10^{-7}}$ |
| $x_0^8(x_0^8 - \frac{1}{2})(x_0^8 - 1) + \frac{1}{10}x_1^{10}$ | $2.39 \times 10^{-4}$ | $6.42 \times 10^{-4}$ | $2.22 \times 10^{-3}$ | $2.22 \times 10^{-3}$ | $\mathbf{7.11 \times 10^{-8}}$ |
| $J_0(20x_0) + x_1^2$ | $1.13 \times 10^{-4}$ | $1.13 \times 10^{-4}$ | $3.70 \times 10^{-3}$ | $6.52 \times 10^{-3}$ | $\mathbf{6.49 \times 10^{-5}}$ |
| $tan(1.4x_0) + tan(1.4x_1)$ | $5.78 \times 10^{-3}$ | $5.78 \times 10^{-3}$ | $6.18 \times 10^{-3}$ | $6.18 \times 10^{-3}$ | $\mathbf{1.37 \times 10^{-6}}$ |
| $log(x_0 + 1.05) + log(x_1 + 1.05)$ | $2.25 \times 10^{-3}$ | $2.25 \times 10^{-3}$ | $3.65 \times 10^{-3}$ | $3.65 \times 10^{-3}$ | $\mathbf{3.86 \times 10^{-6}}$ |
| $\exp(-10(x_0 - 0.1)^2) + \exp(-10(x_1 + 0.1)^2)$ | $2.67 \times 10^{-5}$ | $2.67 \times 10^{-5}$ | $2.47 \times 10^{-4}$ | $3.24 \times 10^{-4}$ | $\mathbf{3.58 \times 10^{-7}}$ |
| $\exp(\sin(x_0) + x_1^2)$ | $1.16 \times 10^{-4}$ | $1.31 \times 10^{-3}$ | $2.56 \times 10^{-3}$ | $3.15 \times 10^{-3}$ | $\mathbf{4.59 \times 10^{-7}}$ |
| $(x_0 + x_1)/(2 + x_0 x_1)$ | $6.97 \times 10^{-5}$ | $6.97 \times 10^{-5}$ | $1.02 \times 10^{-3}$ | $1.44 \times 10^{-3}$ | $\mathbf{1.88 \times 10^{-5}}$ |
| | Time ratio (compared with Legendre-KAN) | | | | |
| $x_0^7 + x_1^4 + x_2^3 + x_3 + 1$ | – | – | – | – | 1.00 |
| $x_0^{15} + x_1^{10} + x_2^5 + x_3 + 1$ | – | – | – | – | 1.00 |
| $x_0^8(x_0^8 - \frac{1}{2})(x_0^8 - 1) + \frac{1}{10}x_1^{10}$ | – | – | – | – | 1.00 |
| $J_0(20x_0) + x_1^2$ | – | – | – | – | 1.00 |
| $tan(1.4x_0) + tan(1.4x_1)$ | – | – | – | – | 1.00 |
| $log(x_0 + 1.05) + log(x_1 + 1.05)$ | – | – | – | – | 1.00 |
| $\exp(-10(x_0 - 0.1)^2) + \exp(-10(x_1 + 0.1)^2)$ | – | – | – | – | 1.00 |
| $\exp(\sin(x_0) + x_1^2)$ | – | – | – | – | 1.00 |
| $(x_0 + x_1)/(2 + x_0 x_1)$ | – | – | – | – | 1.00 |

**The mnist dataset.** We use the shape $[784, 100, 10]$ Legendre-KAN with Adam for 2000 steps on the cross-entropy loss. The max degree of Legendre polynomials is $4$. The result shows that the test accuracy of Legendre-KAN can achieve $98.63\%$. The experimental results may be biased by seeds and initialization. If time permits, we will conduct more tests.

