# OpenReview forum: "Legendre-KAN : High Accuracy KA Network Based on Legendre Polynomials"
_ICLR.cc/2025/Conference — Submitted to ICLR 2025_

### Official Review · Reviewer_8ADV · 2024-10-31

**Soundness:** 2
**Presentation:** 1
**Contribution:** 3
**Rating:** 5
**Confidence:** 4

**Summary:**

The Kolmogorov-Arnold Network (KAN) has been proposed as a significant improvement over MLPs in terms of interpretability and symbolic representation. However, in this paper, researchers have identified issues with the cubic B-spline basis functions used in KAN, specifically their inflexibility due to fixed degrees and knots. As a result, KAN struggles to reduce training error to the precision required for scientific research, leading to mathematical expressions that differ greatly from the true function, thereby limiting its practical applications.

To increase the flexibility of the basis functions in KAN, this paper introduces Legendre-KAN, which employs parameterized Legendre basis functions and normalization layers at KAN's edges. Extensive experiments show that Legendre-KAN achieves 10-100 times greater accuracy than KAN in symbolic representation tasks and in fitting complex nonlinear functions that cannot be easily symbolized. This improvement enhances the accuracy of mathematical relationships within KANs, offering a more effective solution for approximating and analyzing complex nonlinear functions.

**Strengths:**

1. Researchers replaced the B-spline basis functions with Legendre polynomials in KAN, introducing Legendre-KAN. They highlight several benefits of using Legendre polynomials: (1) Legendre polynomials have a global polynomial function approximation space; (2) With higher order, Legendre polynomials can capture more complex patterns and relationships within the data; (3) By applying appropriate orthogonalization, Legendre polynomials are numerical stable.
2. Two technique trick to improve network: (1) Add a normalization layer to normalize the input activation values of each layer to interval $[0,1]$, which unifies the form of each layer's basis function. (2) Use smaller initialization parameters in order to solve the problem of gradient explosion.
3. Experiments in both symbolic representation tasks and fitting complex nonlinear functions that cannot be easily symbolized shows that Legendre-KAN outperforms KAN with 10-100 times greater accuracy.

**Weaknesses:**

1. Lack of detailed and reasonable explanations in section 2.2 "B-spline functions and its fitting characteristic".
	(1) In line 196, figure 3 and figure 4 are used to support the claim "spline functions perform well in smooth regions but may introduce significant errors in certain areas". However, there are two questions. First, figure 3 and figure 4 contradict each other in multiple places: in figure 3(b), Legendre polynomials fits better than B-spline, while in figure 4(b) it is the opposite; in figure 3(c), B-spline fits well, which is inconsistent with figure 4(c). Second, the term "certain areas" is vague, leading to confusion about attributing "significant errors" to the localized fitting caused by piecewise spline functions in the following sentences.
	(2) This part identifies the main drawback that the spline function is localized and piecewise polynomials. However, it is summarized as "the activation functions with lower degrees of freedom prevents Spline-KAN from producing accurate results" in line 254. The conception of "degrees of freedom" and the relationship among "local/global", "order of polynomials" and "degrees of freedom" are confusing.
2. Architecture of Legendre-KAN is missing some key information. It is not clearly specified which module is inherited from KAN and which is modified, or which settings follow KAN and which changes. For instance, in line 352, it is mentioned that "we also combined the best-performing $b(x) = SILU (x)$ with the Legendre basis to enhance the smoothness of the high-order polynomial fitting results." However, it is not clarified the use of SiLU function has already been in the KAN.
3. Some irregularities in the paper writing.
	(1) Lack of definition or repeated definition for the symbols used in the paper. In line 113, a theorem is quoted, but none of the symbol is defined. Meanwhile, same symbols are used in section 3.2 of totally different meanings. Also, in figure 2(a), symbol $B$ with subscript is never defined in the paper.
	(2) Figures and tables lack brief explanations typically provided below them to aid understanding.
	(3) Some writing details such as spelling and punctuation errors. For example, "worse" is spelled as "wrose" in line 257 and a sentence ends with a comma in line 274.

**Questions:**

See Weakness

---

> ### Author Response · Authors · 2024-11-25
> **Response to Reviewer 8ADV**
>
> **Dear Reviewer 8ADV:**
>
> Thank you for your letter and for your comments concerning our manuscript. Those comments are all valuable and very helpful for revising and improving our paper, as well as the important guiding significance to our researches! We have studied comments carefully and have made correction which we hope meet with approval.
>
> **Q1** *Lack of detailed and reasonable explanations in section 2.2 "B-spline functions and its fitting characteristic".*
>
> **A1** Thank you for your positive comments and valuable suggestions to improve the quality of our manuscript. We updated some contents in Section 2 and Section 3. In summary, B-spline performs poorly in the jump regions of the function. Due to its locality and low degree, it lacks degrees of freedom in fitting these regions, which means that the number and degree of the basis functions are lower, and they cannot achieve high-precision fitting of the signal in this region. More importantly, the low fitting precision in the jump regions will affect the overall fitting effect.
>
> **Q2** *Architecture of Legendre-KAN is missing some key information.*
>
> **A2** Thanks for the great suggestions. We describe the complete activation function of Spline-KAN, which will help the reader to compare with the improvement of Legendre-KAN.
>
> **Q3** *Some irregularities in the paper writing.*
>
> **A3** Thank you very much for reading our article carefully! We feel sorry for our poor writings. We tried our best to improve the manuscript and made some changes to the manuscript.
>
> We are very lucky to have met a responsible reviewer. Thank you very much for your description of the strengths of the manuscript. If there are any other problems, we will try our best to solve them. In addition, comparative experiments with other KANs are in progress. If time permits, we will add relevant experimental results in the appendix D.

---

> > ### Comment · Reviewer_8ADV · 2024-12-02
> >
> > After reading the responses and all the reviews, I will keep my score.

---

### Official Review · Reviewer_Y68h · 2024-11-01

**Soundness:** 2
**Presentation:** 2
**Contribution:** 2
**Rating:** 3
**Confidence:** 3

**Summary:**

This paper proposes Legendre-KAN, a variant of the Kolmogorov-Arnold Network (KAN) that integrates Legendre polynomials into the activation functions to improve performance. By utilizing Legendre basis functions, along with skills such as normalization and reduced initialization parameters, the authors aim to enhance the accuracy and parameter efficiency of KAN in symbolic representation tasks. Empirical evaluations demonstrate that Legendre-KAN achieves better fitting accuracy compared to the standard KAN.

**Strengths:**

•	Incorporating Legendre basis functions into KAN is a novel approach that appears to contribute positively to the network's performance.
•	The empirical results seem to show that Legendre-KAN results in improved fitting accuracy.

**Weaknesses:**

•	The paper lacks a theoretical analysis to substantiate why Legendre polynomials outperform B-spline functions.
•	There are grammatical errors and incomplete sentences in the manuscript, notably in Section 3.1, which impede comprehension.
•	The evaluation is confined to symbolic representation tasks. While recent study[1] shows that KAN is found to be better than MLP only in symbolic formula representation, but still inferior to MLP on other tasks such as machine learning, CV, NLP and audio processing. It would be more convincing if the proposed method with KAN and MLP could be tested on tasks other than symbolic representation.
•	The structure of the paper is somewhat disorganized, with experimental results appearing in sections typically reserved for background and methodology.

[1] Yu, Runpeng, Weihao Yu, and Xinchao Wang. "Kan or mlp: A fairer comparison." arXiv preprint arXiv:2407.16674 (2024).

**Questions:**

•	A deeper theoretical comparison of Legendre polynomials and B-spline functions is necessary to strengthen the argument for the proposed method.
•	Test the proposed method on additional tasks beyond symbolic representation to demonstrate its effectiveness in other domains and strengthen the overall claims.
•	Improve the clarity and grammatical correctness of the writing to better convey the proposed method's details and implications.
•	Enhance the descriptions in figure captions for clearer understanding.
•	Enhance the clarity of the figures, especially Figure 6, and ensure that all labels and legends are accurate. For instance, the labels in Figure 4b appear to be reversed.

---

> ### Author Response · Authors · 2024-11-27
> **Response to Reviewer Y68h**
>
> Thank you for your suggestions.
>
> # QA
>
> **Q1** * A deeper theoretical comparison of Legendre polynomials and B-spline functions is necessary to strengthen the argument for the proposed method.*
>
> **A1** Thank you for your valuable advice. We provide detailed experimental proof as to why Legendre-KAN is better than Spline-KAN. This may help to understand.
>
> **Q2** *• Test the proposed method on additional tasks beyond symbolic representation to demonstrate its effectiveness in other domains and strengthen the overall claims.*
>
> **A2** Thanks for your review! We are already hard at work experimenting. However, due to equipment and time constraints, we currently only conduct experiments on the MNIST data set. In the future, we will conduct additional experiments to strengthen our claims.
>
> **Q3/4/5** *• There are grammatical errors and incomplete sentences in the manuscript •Enhance the descriptions in figure captions for clearer understanding. • Enhance the clarity of the figures, especially Figure 6, and ensure that all labels and legends are accurate. For instance, the labels in Figure 4b appear to be reversed.*
>
> **A3** Thank you for your careful reading! We are sorry for these errors. In the latest version, we have added explanations under the figures and updated Figure 4b. Due to tool limitations, the clarity of some figures may still be poor.

---

### Official Review · Reviewer_8ZCw · 2024-11-03

**Soundness:** 2
**Presentation:** 2
**Contribution:** 2
**Rating:** 3
**Confidence:** 2

**Summary:**

This paper proposes a new variant of the Kolmogorov-Arnold Network (KAN) called Legendre-KAN, designed to improve the accuracy of symbolic representation and function approximation. The main advancement here is the replacement of the traditional B-spline basis functions in KANs with parameterized Legendre polynomials, which offer higher degrees of freedom and are known for their global approximation capabilities and numerical stability.

**Strengths:**

The shift from B-splines to Legendre polynomials appears well-motivated, and the results convincingly show an increase in accuracy for small problems.

**Weaknesses:**

1. There have been several KAN alternatives proposed at this point -- Fourier KANs, Wavelet KANs, RBF KANs, etc. There are no comparisons to those alternatives.
2. The paper starts with mention of areas which require high accuracy and precision, however the target experiments are extremely small scale. Even the "complex" nonlinear functions are simple polynomials where no-one uses neural networks for approximation.

In the current state, the paper requires significant revision before it can be considered for publication.

**Questions:**

See limitations for the main issues with the paper.

---

> ### Author Response · Authors · 2024-11-25
> **Response to Reviewer 8ZCw**
>
> Dear Reviewer 8ZCw:
>
> Thank you for your comments concerning our manuscript. Thank you for your time and effort in reviewing the manuscript. The explanation of Legendre-KAN's advantage in high-precision fitting has been revised in Sections 2.2 and 3.1. We are sorry for the careless writing in English.
>
> **Q1** *There have been several KAN alternatives proposed at this point -- Fourier KANs, Wavelet KANs, RBF KANs, etc. There are no comparisons to those alternatives.*
>
> **A1** Thank you very much for your advice! We will add experiments comparing with other KAN in the next edition, expected in a few hours. It seems that Legendre-KAN performs better than other KAN in symbolic experiments, which may be superior to Legendre's global polynomials versus his orthogonality. The code for comparison comes from the github release. There seemed to be some problems with the parameters of wav-kan, and we adopted the torch.rand() to improve its accuracy. If time permits, we will conduct more comparative experiments.
>
> [1] Xu J, Chen Z, Li J, et al. FourierKAN-GCF: Fourier Kolmogorov-Arnold Network--An Effective and Efficient Feature Transformation for Graph Collaborative Filtering[J]. arXiv preprint arXiv:2406.01034, 2024.
>
> [2] Bozorgasl Z, Chen H. Wav-kan: Wavelet kolmogorov-arnold networks[J]. arXiv preprint arXiv:2405.12832, 2024.
>
> **Q2** *The paper starts with mention of areas which require high accuracy and precision, however the target experiments are extremely small scale. Even the "complex" nonlinear functions are simple polynomials where no-one uses neural networks for approximation.*
>
> **A2** Thank you again for your positive comments and valuable suggestions to improve the quality of our manuscript. Appendix C has some experiments with complex functions. We are conducting some other experiments on the Feynman dataset and will add the results to the appendix if time permits. We would like to emphasize that Legendre-KAN has a very amazing accuracy on the polynomial part of signals or formulas, and the polynomial part exists in a large number of scientific studies, including quantum physics, analytical and computational chemistry. More importantly, in some symbolic regression tasks, the polynomial part of the signal is difficult to represent. In KAN, accurate polynomial results cannot be obtained by matching the optimal function, but Legendre-KAN can accurately express this type of activation function. In addition, Legendre-KAN still shows high accuracy in some non-polynomial parts of the signal, such as $sin(x)$ or the Bessel function, which is closely related to the high degree of freedom and orthogonality of Legendre polynomials and our improvement of the network.

---

### Official Review · Reviewer_5qh9 · 2024-11-03

**Soundness:** 1
**Presentation:** 3
**Contribution:** 1
**Rating:** 3
**Confidence:** 2

**Summary:**

The authors introduce a new network called Legendre-KAN, which combines Legendre polynomials with the Kolmogorov-Arnold (KA) theorem. The motivation behind this development is that the standard Kolmogorov-Arnold Network (KAN), due to the inherent limitations of B-splines, cannot sufficiently reduce training error for complex tasks such as solving partial differential equations. The authors conduct a series of experiments in the field of symbolic regression to demonstrate that their approach outperforms both KAN and MLP in terms of test loss when fitting the equations.

**Strengths:**

- The Legendre-KAN achieves lower test set losses on a set of symbolic expressions compared to the standard KAN.
- The authors provide an extensive overview of the method.

**Weaknesses:**

- The current results are limited to only a small set of equations. In the field of symbolic regression, there are well-established benchmarks, such as SRBench and SRSD, on which novel approaches are typically tested. The authors should test on these benchmarks.
- No analysis with noise is performed.
- It is unclear what scientific contribution this approach brings. If the authors present this work as a contribution to symbolic regression, they should test it against strong and well-established baselines (such as Operon, DSR, and Neural Symbolic Regressors) rather than only KAN. Alternatively, if the authors are focusing solely on the KAN comparison, they should explain why they chose the symbolic regression task and what better performance on this task implies.

**Questions:**

- Do you have any performance improvements in non-symbolic regression tasks compared to KAN? For example, in the abstract, you mention solving partial differential equations, and in your related work, you mention approaches where KAN has been used in the context of Graph Neural Networks and Transformers. Did you test your approach on these tasks and settings?
- What do the terms “time ratio,” “Equal params,” “best k/grid,” and “best k” mean in the tables?

---

> ### Author Response · Authors · 2024-11-25
> **Response to Reviewer 5qh9**
>
> Thank you very much for your instructive suggestions and questions! Your suggestion has inspired us a lot.
>
> For **weakness1**:
>
> We would like to quote the reply of **KAN**(https://openreview.net/forum?id=Ozo7qJ5vZi) :
>
> Regarding comparison to symbolic regression methods: KAN, as a network-based method, has strong capability (in fitting even non-symbolic functions) that makes it unfavorable for standard symbolic regression benchmarks. For example, KAN ranks second-to-last in GEOBENCH (https://openreview.net/forum?id=TqzNI4v9DT), whereas the last-ranked one EQL is also a network-based model, which turned out to be useful at least for certain problems despite its inability to do well on benchmarks. On the one hand, we would like to explore ways to restrict KANs' hypothesis space so that KANs can achieve good performance on symbolic regression benchmarks. On the other hand, we want to point out that KANs have good features that are hard to evaluate with existing benchmarks: (1) interactivity. It is very hard to "debug" evolutionary-based symbolic regression methods since the evolution process is long and random. However, it is relatively easier to visualize the training dynamics of KANs, which gives human users intuition on what could go wrong. (2) The ability to ``discover'' new functions. Since most symbolic regression methods require the input of the symbolic library, they cannot discover things they are not given. For example, if the ground truth formula contains a special function but is not given in the symbolic library, all SR methods will fail definitely. However, KANs can discover the need for a new function whose numerical behavior suggests maybe it is a Bessel function; see Figure 23 (d) for an example.
>
> Further, we want to emphasize that Legendre-KAN can fit those parts of polynomials accurately that cannot be represented by symbols and ubiquitous in scientific research, and even give their expressions.
>
> For **weakness2**:
>
> As suggested by the referee, we have tried our best to verify Legendre-KAN’s ability in other tasks. It is a shame that we do not have enough time to complete all of those tasks.
>
> # Q/A
>
> **Q1** *Do you have any performance improvements in non-symbolic regression tasks compared to KAN?*
>
> **A1** Thank you for your sincere advice! We're doing the best we can with the experiment. Comparative experiments for applications such as differential equations are ongoing.
>
> **Q2** *What do the terms “time ratio,” “Equal params,” “best k/grid,” and “best k” mean in the tables?*
>
> **A2** Thank you for your question. We apologize for the lack of explanation on some indicators.
>
> All the $\textbf{ratios}$ represent the ratio of the indicator of a certain network to this indicator of Legendre.
>
> For each tasks, we test the network with different parameter quantity.
> For each tasks, we test the network with different parameter quantity.
> The parameter quantity for the lowest error of the result of
> the Legendre-KAN is assumed to be $k_l$.
> We divide all the results of the of networks into two parts.
> The $\textbf{first part}$ is the part where the parameter quantity is
> less than or equal to or slightly greater than $k_l$.
> The $\textbf{other part}$ is the part where the number of parameters
> is less than or equal to the maximum number of basis functions specified.
> The former is to compare the situation when the two networks' parameter quantities are equal. The second part is used to compare the optimal fitting accuracy of the network under a certain number of parameters. For the result with the lowest error in the first part, the number of basis functions is described as $\textbf{Equal params}$. For the result with the lowest error in the second part, the number of basis functions is described as $\textbf{best k/grid}$ in Spline-KAN and $\textbf{best k}$ in other networks.

---

### Meta-Review · Area_Chair_TpZP · 2024-12-18

**Metareview:**

The paper proposes Legendre-KAN, a variant of the recently proposed Kolmogorov-Arnold Network (KAN) architecture. The proposed method replaces the cubic B-spline basis functions with parameterized Legendre polynomials. This modification aims to enhance the flexibility and accuracy of KANs, particularly in tasks involving symbolic regression and function approximation. The authors highlight the potential advantages of Legendre polynomials to address limitations in B-splines such as fixed degree and localized fitting. Experimental results suggest that Legendre-KAN achieves improvements in fitting accuracy compared to standard KANs.

The paper has several notable weaknesses that limit its overall contribution. A concern is the limited scope of the experiments, which are confined to a small set of equations and do not include established benchmarks (eg. SRBench, SRSD, etc..). Additionally, no meaningful analysis/comparison against well-established symbolic regression baselines is provided. While the work aims to improve KANs, it does not address comparisons with many other KAN variants that have recently been proposed (Fourier-KAN, Wavelet-KAN, and several others) leaving its relative advantages unclear. Furthermore, the theoretical explanations have been judged insufficient by the reviewers.

For the reasons outlined above, the panel of reviewers has decided not to accept the paper in its current form. While the idea of leveraging Legendre polynomials in KANs shows significant potential, the paper as it stands lacks the empirical validation and theoretical depth required for acceptance at ICLR. The panel strongly encourages the authors to address these weaknesses in a future submission, specifically by testing on established benchmarks, conducting more comprehensive comparisons, and enhancing the clarity and organization of the manuscript.

**Additional Comments On Reviewer Discussion:**

The reviewers requested several clarifications, which the authors addressed to some extent. However, the reviewers also noted that the experiments were too limited in scope. While the authors acknowledged this feedback and indicated they were working on expanding the experiments, they cited constraints in time and computational resources as reasons for the incomplete results.

---

### Decision · Program_Chairs · 2025-01-22

Reject